# The ABCs of Antigen Presentation by Stromal Non-Professional Antigen-Presenting Cells

**DOI:** 10.3390/ijms23010137

**Published:** 2021-12-23

**Authors:** Tom J. Harryvan, Sabine de Lange, Lukas J.A.C. Hawinkels, Els M.E. Verdegaal

**Affiliations:** 1Department of Gastroenterology & Hepatology, Leiden University Medical Center, 2333 ZA Leiden, The Netherlands; sabine.delange@gmail.com; 2Department of Medical Oncology, Oncode Institute, Leiden University Medical Center, 2333 ZA Leiden, The Netherlands

**Keywords:** nonprofessional antigen presentation, stromal immunobiology, immuno-oncology

## Abstract

Professional antigen-presenting cells (APCs), such as dendritic cells and macrophages, are known for their ability to present exogenous antigens to T cells. However, many other cell types, including endothelial cells, fibroblasts, and lymph node stromal cells, are also capable of presenting exogenous antigens to either CD8+ or CD4+ T cells via cross-presentation or major histocompatibility complex (MHC) class II-mediated presentation, respectively. Antigen presentation by these stromal nonprofessional APCs differentially affect T cell function, depending on the type of cells that present the antigen, as well as the local (inflammatory) micro-environment. It has been recently appreciated that nonprofessional APCs can, as such, orchestrate immunity against pathogens, tumor survival, or rejection, and aid in the progression of various auto-immune pathologies. Therefore, the interest for these nonprofessional APCs is growing as they might be an important target for enhancing various immunotherapies. In this review, the different nonprofessional APCs are discussed, as well as their functional consequences on the T cell response, with a focus on immuno-oncology.

## 1. Introduction

Antigen presentation is a crucial aspect in mounting an effective immune response against pathogens and tumors. Professional antigen-presenting cells (APCs), such as dendritic cells (DCs), constantly scan tissues for antigens, including pathogen-derived, as well as aberrantly expressed, antigens (neo-antigens) resulting from malignant transformation. Upon the uptake of exogenous antigen and concomitant activation via pathogen or danger/damage-associated molecular pattern receptors, DCs differentiate into mature DCs, defined by the upregulated expression of co-stimulatory molecules, and travel to the lymph nodes [1]. There, they subsequently prime T cells and initiate an immune response. The net outcome on T cell function is dependent on the type of co-regulatory signals provided by the DC, including co-stimulatory signals (e.g., CD80/86) and co-inhibitory signals (e.g., PD-L1). A lack of sufficient co-stimulatory signals leads to the induction of anergy and thus promotes immunological tolerance, while strong co-stimulatory signals license T cells to become fully activated [2]. Next to DCs, B cells and macrophages are also considered professional APCs and their role in T cell immunity is firmly established. Other cell types have been recognized to also possess the capacity for exogenous antigen presentation. These cell types are collectively referred to as nonprofessional, or “amateur”, APCs [3]. Both professional and nonprofessional APCs are able to take up exogenous antigens and present them in both classes of major histocompatibility complex (MHC) molecules: class I and class II (Figure 1).

MHC class I (MHCI) is expressed by all nucleated cells in the body and presents peptides on the cell surface, which are recognized by CD8+ T cells. Cells continuously present endogenous antigens in MHCI, which will normally only lead to the activation of T cells specific for “non-self” antigens resulting from infection or malignant transformation. On the contrary, MHC class II (MHCII) is primarily found on professional APCs and is used almost exclusively for the presentation of exogenous antigens to CD4+ T cells.

However, the process of antigen presentation is not as rigid as described above, as exogenous antigens can also be presented in MHC class I molecules. This process is called cross-presentation and results in the activation of CD8+ T cells. Cross-presentation is crucial in initiating the immune response against viruses that have infected cells other than DCs, in the generation of peripheral tolerance against self-antigens, but also in the generation of an adaptive immune response against tumors [4]. Nonprofessional APCs can present exogenous antigens on both MHCI and MHCII and thus influence both CD8+ and CD4+ T cell function, although MHCII is usually only expressed under inflammatory conditions [5].

The process of exogenous antigen processing does differ between the two MHC classes. For exogenous antigen presentation in MHCI, the acquired antigen can be processed via two different pathways: the vacuolar pathway and the endosome-to-cytosol pathway (Figure 1**,** left panel) [4,6,7,8]. When processed via the vacuolar pathway, the internalized antigen is degraded by lysosomal proteases in endosomes, including cathepsins, and subsequently loaded on recycling MHCI molecules. Processing via the endosome-to-cytosol pathway involves the transportation of exogenous antigens to the cytosol for degradation by the proteasome. Subsequently, the antigens are transported into the endoplasmic reticulum (ER) by the transporter-associated with antigen processing (TAP) complex. In the ER, the peptide is loaded in MHC class I molecules with the aid of the peptide-loading complex (PLC). The PLC consists of the molecules calnexin, calreticulin, ERp57, B2M, and tapasin and facilitates binding of the peptide in the antigen-binding cleft of MHCI. Once the peptide is bound, the MHCI-peptide complex is released from the peptide-loading complex, exits the ER and is transported to the cell surface [9].

For presentation of exogenous antigens via MHCII, the protein is taken up and cleaved by cathepsins and other proteases in endosomal compartments. The MHCII molecule is formed in the ER with the aid of the chaperone calnexin and after binding of the invariant chain (Ii) to the MHCII molecule, exported to the endolysosomal compartment. Here, Ii is cleaved by proteases, leaving the class II-associated invariant chain peptide (CLIP) in the binding pocket, which prevents premature loading of peptides. CLIP is released by HLA-DM and thus enables binding of the exogenous, processed antigen after which the MHCII–peptide complex will be transported to the cell surface (Figure 1, right panel) [10]. 

The molecular process of antigen presentation is very dynamic and the expression of many of the proteins discussed above is regulated by inflammatory cytokines, such as interferon gamma (IFNγ). This suggests that changes in the cellular environment during, for example, infection or tumor growth can also potentially endow nonprofessional APCs with enhanced antigen processing capacity and thus affect T cell activation [11,12].

In this review, we discuss the current literature on nonprofessional APCs, as well as their effect on T cell function, with a focus on stromal cells and immuno-oncology. The role of nonprofessional APCs in other diseases has been expertly reviewed elsewhere [13,14].

## 2. Nonprofessional Antigen Presentation in the (Tumor) Stroma

Stromal cells are nonepithelial cells that include fibroblasts, endothelial cells (ECs), smooth-muscle cells (SMC), and leukocytes, which fulfil an important part in tissue homeostasis. Expansion of the stromal compartment is seen in a variety of pathologies, including inflammation, but is most striking in cancer where the stroma can reach up to 80% of the tumor content [15]. In many tumors, (tumor-specific) T cells are thus closely juxtaposed to stromal cells, giving rise to distinct T cell–stromal cell interactions. Current research has mostly focused on indirect ways of stromal cell-mediated T cell interactions, e.g., cytokine-mediated, and has been reviewed elsewhere [16,17]. Here, we discuss the direct interaction between the major stromal cell subtypes with antigen-presenting capacities and T cells, under physiological and pathological conditions, as well as their effects on T cell function.

### 2.1. (Cancer-Associated) Fibroblasts

As the most common connective tissue cells in our body, fibroblasts are best known for their ability to model the extracellular matrix (ECM), and play important roles in a variety of pathologies ranging from inflammation [18] to oncology [19] via interactions with surrounding epithelial and immune cells. Recent work using single-cell technologies has revealed a variety of fibroblast subsets [20,21,22,23,24], including those with features of antigen presentation capacity. These so-called antigen-presenting fibroblasts, which express MHCII, were first identified in pancreatic ductal adenocarcinoma (PDAC) [25]. This plethora of data has complicated fibroblast nomenclature, reviewed elsewhere [26], but here, we mostly describe functional aspects of fibroblast-mediated antigen presentation and, thus, use the generic term, (cancer-associated) fibroblast, for clarity. Below, we discuss the role of fibroblasts in antigen presentation under physiological and pathological conditions, as well as the functional consequences on immune cell activation.

#### 2.1.1. Nonprofessional Antigen Presentation by Fibroblasts under Physiological Conditions

Under physiological conditions, antigen-presenting fibroblast subsets have been identified in the human lung [27], liver [28,29], and colon [30,31] by their expression of MHCII molecules. Intriguingly, these organs are in (in)direct connection with extensive microbial communities and it is thus tempting to attribute a role for these fibroblasts in maintaining immunological tolerance. Evidence for this hypothesis is provided by fibroblasts in the human lung, which were shown to lack expression of the co-stimulatory molecules CD80 and CD86, but did express 4-1BBL, OX-40L, and CD70 [27]. This suggests that these fibroblasts are able to activate memory T cells but are not capable of T cell priming. These fibroblasts were indeed shown to activate antigen-specific memory T cells after exposure to IFNγ, mimicking a local inflammatory environment, through the MHC-II-mediated antigen presentation of lung-resident bacteria, resulting in IFNγ and interleukin (IL)-17a production [27]. In the human colon, a similar subset of fibroblasts has been identified [30,31] that is characterized by the expression of alpha smooth muscle actin (αSMA) and MHCII. However, it is currently unknown whether this particular subset is also capable of exogenous antigen presentation and thus can be considered a “true” nonprofessional APC. In this regard, it is interesting to note that these fibroblasts do express CD80 and CD86 and therefore might also play a role in T cell priming. This type of co-stimulatory molecule expression is also found in resident, murine hepatic fibroblasts, also called hepatic stellate cells (HSCs), which were shown to be able to present an exogenous model antigen, ovalbumin (OVA), in both MHCI and MHCII molecules [29]. As expected, these fibroblasts could also induce antigen-specific proliferation of ovalbumin-specific CD8+ and CD4+ T cells, respectively. However, compared to professional liver-resident APCs (Kupffer cells), the induction of proliferation was significantly lower, indicating suboptimal T cell activation by the HSCs. Detailed assessment revealed that this was not attributable to a difference in antigen processing capacity between professional and nonprofessional APCs, but rather to a difference in expression of co-stimulatory molecules. Collectively, these studies show that the exogenous antigen (cross)presentation machinery is present in a variety of fibroblasts and that the extent of T cell activation that subsequently occurs might not be a result of insufficient antigen processing but rather the absence of the appropriate co-stimulatory cues. As the expression of these co-stimulatory molecules is dynamic and strongly regulated via inflammatory mediators, such as IFNγ, it is thus conceivable that the capacity of fibroblasts (and other nonprofessional APCs) to mediate the T cell response depends on the (tumor) micro-environment in which the antigen presentation occurs.

#### 2.1.2. Nonprofessional Antigen Presentation by Fibroblasts under Pathological Conditions and the Effect of the Local Micro-Environment

The capacity of fibroblasts to present exogenous antigens has recently gained renewed interest, particularly in cancer. As the main cellular constituent of most solid tumors, cancer-associated fibroblasts (CAFs) are currently intensively investigated with regard to their role in regulating the anti-tumor T cell response [16]. Interestingly, CAFs were shown to be capable of directly influencing the anti-tumor T cell response, by the induction of apoptosis of tumor-specific T cells in a murine model of melanoma [32]. The uptake and processing of antigens was shown to be more efficient in CAFs, compared to normal fibroblasts, and resulted in the PD-L2- and FASL-mediated induction of T cell apoptosis. These findings show that CAFs differ from normal fibroblasts with respect to antigen processing but also with respect to the net effect on T cell function. It is currently unknown which components of the antigen processing machinery are responsible for the enhanced antigen processing observed in CAFs and to what extent antigen presentation by CAFs contributes to the net regulation of the anti-tumor T cell response. Recent data from our group (Harryvan et al., in revision) showed that in human colorectal cancer (CRC), in vitro-cultured CAFs acquire enhanced cross-presentation through upregulation of the lysosomal protease cathepsin S, a key component of the vacuolar pathway. A subset of primary CRC-derived CAFs showed upregulation of cathepsin S in vitro, compared to their matched normal counterparts. This effect could be mimicked by the exposure of normal fibroblasts to CRC-conditioned medium, implicating a currently unknown tumor-derived factor to be responsible for this upregulation. Importantly, antigen presentation by these tumor-conditioned fibroblasts resulted in diminished cytotoxic T cell function and was accompanied by a decrease in activating (CD137) and increase in inhibitory (TIM3, LAG3, and CD39) checkpoint molecule expression.

Findings from the studies above relied on murine models and mostly in vitro human findings, so it is currently unclear what the effect of CAF-mediated exogenous antigen presentation is in human cancer in vivo. Recent studies aimed at exploring the CAF heterogeneity in human tumors have revealed CAF subsets that possess antigen-presenting features. In PDAC, a subpopulation of CAFs has been identified with single-cell RNA sequencing that expresses MHCII [25]. This study also showed that CAFs that express MHCII are a dynamic population that can upregulate myofibrotic CAF markers, showing the plasticity of CAFs, and reaffirm other findings showing that environmental cues dictate their final antigen-presenting capacity [32]. Isolation of these antigen presenting CAFs was feasible and in vitro antigen presentation assays showed the ability of these CAFs to present MHCII-restricted antigens to CD4+ T cells [25]. However, these CAFs did not express CD80, indicating that their antigen presentation might result in T cell anergy induction. Of note, MHCII expression is a clear necessity for the presentation of exogenous antigens to CD4+ T cells and is, therefore, a suitable marker to identify potential nonprofessional APC-CAF subsets. However, such a discriminating marker for exogenous antigen presentation via cross-presentation is lacking as all cells express MHCI, and therefore hampers the identification of cross-presenting fibroblast subsets. Insights into differences in expression of the antigen processing machinery components, such as the expression of lysosomal proteases and (immuno)proteasome subunits, between CAFs and normal fibroblasts could aid in identification of these subsets.

Finally, findings from other research areas such as auto-immune diseases might also help in elucidating the requirements for efficient fibroblast-dependent antigen presentation and the subsequent effect on T cell function. For example, antigen-presenting synovial fibroblasts from rheumatoid arthritis patients were shown to be able to activate CD4+ T cells to release IL-2 [33]. Moreover, mesenchymal stromal cells (MSCs), a cell type that has been implied to contribute to the CAF populations found in several cancers [34,35,36], have received attention as a cellular therapeutic in multiple auto-immune diseases [37] due to their immune regulatory/anti-inflammatory properties. MSCs are characterized by the expression of molecules including CD90, CD105, and Stro-1 and are able to differentiate into multiple lineages, including osteoblasts, chondrocytes, and adipocytes [38,39,40,41]. Interestingly, MSCs exposed to IFNγ were shown to be able to internalize a soluble antigen via endocytosis, process it, and load it onto MHCII. Subsequently, these cells were able to prime CD4+ T cells, due to the IFNγ-mediated upregulation of CD80, resulting in the production of effector molecules such as IL-2. Another study also provided direct evidence for cross-presentation of the model antigen OVA in MSCs both in vitro and in vivo [42]. Finally, an elegant study showing the conditional functioning of MSCs as nonprofessional APCs showed that the pre-treatment of mice with ovalbumin-pulsed, IFNγ-treated MSCs provided complete protection after subsequent challenge with ovalbumin-expressing E.G7 tumors [43]. Combined, these studies show that MSCs can function as potent nonprofessional APCs and it would, therefore, be of considerable interest to see whether MSC-derived CAFs retain these features or whether currently identified antigen-presenting CAF subsets are of MSC origin.

Altogether, the studies described above reveal the potential of (cancer-associated) fibroblasts to sample and present antigens from their surroundings to T cells, activating and, in some cases, even priming antigen-specific T cells. However, the lack of appropriate co-stimulation (e.g., CD80/CD86) often results in the induction of anergy, thereby promoting immunological tolerance. Under physiological conditions and in auto-immune diseases, this can be a beneficial outcome by preventing an unwanted T cell response against microbial or self-antigens. In oncology, this can, however, lead to the suppression of tumor-specific T cells and thus promote tumor growth. The net effect of fibroblast-mediated exogenous antigen presentation on T cell function is thus the combination of the innate ability of fibroblasts to present exogenous antigens, combined with the cues from the micro-environment. These cues can conditionally enhance fibroblast antigen-presenting abilities and/or alter the expression of co-stimulatory molecules that ultimately define the resulting T cell response.

### 2.2. Lymph Node Stromal Cells

Secondary lymphoid organs, including lymph nodes and the spleen, are important sites for the activation of circulating T cells and are thus directly involved in regulating T cell responses against infections and tumor cells [44]. These structures contain multiple cell types, including those of stromal origin. Lymph node stromal cells (LNSCs) consist of lymphatic endothelial cells (LECs), blood endothelial cells (BECs), and fibroblastic reticular cells (FRCs) [45]. These specialized subsets of cells are potent nonprofessional APCs and have been shown to fulfil crucial roles in regulating the T cell response, reviewed extensively elsewhere [45,46,47]. Located strategically at sites of T cell priming, LNSCs maintain peripheral tolerance by expressing genes encoding for peripheral tissue antigens to prevent the expansion of self-reactive T cells. Due to their lack of appropriate co-stimulatory molecules, this form of antigen presentation leads to anergy induction in CD8+ T cells, in part mediated through PD-L1, ultimately leading to tolerance toward self-antigens [46,48]. LNSCs have a low expression of MHCII under physiological conditions but they can also acquire these molecules directly from DCs present in the lymph node through a process called trogocytosis. In this process, MHCII–peptide complexes are transferred from the cell surface of DCs to LNSCs. After the transfer of the MHCII–peptide complex, LECs can present the MHCII–peptide complex to CD4+ T cells, resulting in the apoptosis of CD4+ T cells, in a PD-L1 dependent manner [49]. Direct processing and presentation in MHCII by LNSCs has also been described and again illustrates the conditional functioning as nonprofessional APC given the right cue. Under physiological conditions, MHCII expression in LNSCs is generally low, but in a model of vaccinia virus infection, LNSCs acquired enhanced MHCII-mediated antigen presentation capacity, and this enabled them to contract the developing CD4+ T cell expansion [50].

Besides acquiring peptide–MHC II complexes directly from their environment, LNSCs are also able to take up exogenous antigens and cross-present these in MHCI molecules. The LEC-dependent cross-presentation of exogenous antigens was shown to result in the proliferation of CD8+ T cells [51]. However, these T cells produce less IFNγ, and increased apoptosis of T cells was observed, which indicates that LECs induce immune tolerance, in line with the net effect observed after the presentation of endogenous proteins [46,48]. Mechanistically, PD-L1 was found to be upregulated, offering an explanation for the observed tolerance phenotype. In relation to cancer, LNSCs have been shown to be able to present tumor-derived antigens via cross-presentation, leading to immune evasion of tumors, by inducing tolerance in tumor-specific T cells [47,52]. However, another study revealed that not all T cells that are cross-primed by LECs enter apoptosis or become anergic, as a small portion of T cells can form a long-lived memory T cell population that converts to effector T cells upon antigen exposure [51]. The exact mechanisms that underlie these opposing effects are currently not well understood but crucial to be able to avoid tolerance induction and instead potentiate tumor-specific T cells.

In summary, LNSCs are spatially located at a prime site of immune regulation and current research singles these cells out as key regulators of T cell function. Overall, their role seems to involve the maintenance of peripheral tolerance by the induction of T cell anergy under physiological conditions, and this process can be hijacked in cancer by suppressing the establishment of an effective anti-tumor T cell response.

### 2.3. Endothelial Cells

Another major stromal cell type that possesses antigen-presenting capacity are endothelial cells (ECs). These cells form the inner lining of blood vessels and thus form a direct barrier between blood and the tissue, which T cells have to pass to find their target. As such, ECs are perfectly situated to present blood-borne antigens, as well as antigens acquired from adjacent tissue, to passing T cells. During tumor development, the formation of new blood vessels, via the recruitment of ECs from surrounding vessels, is a crucial step to provide sufficient nutrients and oxygen to the tumor. However, this also enables subsequent T cell infiltration, highlighting the need for a deeper understanding of EC-mediated antigen presentation in immuno-oncology.

Under physiological conditions, liver sinusoidal ECs (LSECs) have been intensively studied and shown to be able to cross-present soluble exogenous antigens very efficiently using similar mechanisms as DCs. However, the net effect on T cell function is completely opposite to DCs, being the induction of tolerance, in a PD-L1-dependent manner [53,54]. Additionally, LSECs can present exogenous antigens in MHC class II molecules to CD4+ T cells and, similar to CD8+ T cells, this also results in tolerance induction [55,56]. Due to their potent function as nonprofessional APC, the kinetics of antigen presentation of LSECs in comparison to DCs has been studied more in depth. Unexpectedly, LSECs were shown to need lower antigen concentration to cross-prime CD8+ T cells compared to professional APCs. However, the lifetime of processed antigens was considerably shorter in LSECs, and this might affect the timespan in which it can affect T cell function [57]. Moreover, the tolerogenic effect imposed by LSECs can be overruled by the presence of exogenous IL-2, potentiating cytotoxic T cell function [58]. This emphasizes the need for appropriate co-stimulation and the influence of the micro-environment in determining the net result of nonprofessional antigen presentation. This is further demonstrated under pathological conditions, such as in the context of a viral infection. Antigen presentation by LSECs in such a highly inflammatory micro-environment can result in the formation of a memory-like T cell population that can be reactivated upon antigen exposure in the context of a subsequent inflammation [59,60]. A similar phenomenon has also been observed in the context of autoimmunity. In diabetes, beta cell-produced insulin can be endocytosed by ECs in the pancreatic islets and cross-presented to CD8+ T cells. This does not lead to tolerance induction, but instead potentiates CD8+ T cell function, ultimately leading to destruction of the beta cells of the pancreas [61]. In contrast, this hyperinflammatory environment is lacking in some tumors where LSECs have been shown to take up tumor-derived cell fragments and soluble antigens, which results in CD8+ tolerance toward tumor cells [62,63]. To summarize, the function of endothelial cells is not only facilitating T cell entrance to the tissue but also directly modulating T cell function by antigen presentation of both tissue-derived, as well as (blood-derived), soluble exogenous antigens, in which the inflammatory cues provided can steer the final T cell response from anergic to activated.

## 3. Summary

Combined, the studies on the major stromal subtypes mentioned above show similarities, in that all these cells possess an innate capacity for antigen presentation and that under physiological conditions, this is predominantly used to aid in maintaining peripheral tolerance. However, under pathological conditions, the antigen presentation capacity is enhanced and the net effect on T cell function also becomes more strongly regulated by specific cues derived from the micro-environment. In oncology (Figure 2A), the tolerogenic effects of nonprofessional APCs are detrimental for an effective anti-tumor T cell response but, given the data collected in the fields of virology and auto-immunity, there is concrete evidence that increasing the amount of inflammation in the micro-environment can also result in the potentiation of T cells by nonprofessional APCs (Figure 2B). Of note, smaller stromal subsets including smooth muscle cells, granulocytes, and certain T cell subsets also possess the capacity to act as nonprofessional APCs, but the available data are limited so far, which complicates functional assessment at this time.

## 4. Conclusions

For several decades, it has been known that exogenous antigen presentation is not merely restricted to professional APCs but that multiple other cell types also possess this ability. These nonprofessional APCs are mostly stromal cell subtypes, of which (cancer-associated) fibroblasts, LNSCs, and ECs have been most intensively studied. Recent insights in stromal immunobiology [23,64,65] reveal an important role for regulation of the T cell response by different stromal cell types, but it is largely unknown how nonprofessional antigen presentation directly contributes to regulating T cell responses. The assessment of the differences between nonprofessional APCs and professional APCs shows that the uptake and processing routes of exogenous antigens are generally similar, with notable exceptions for the antigen half-life and the expression of MHCII. Under physiological conditions, MHCII expression is absent on the majority of nonprofessional APCs, although some exceptions have been reported [31]. In different pathological states, MHCII is expressed on a larger subset of nonprofessional APCs, mainly driven by IFNγ signaling, and this extends their capacity to also modulate CD4+ T cell responses. Antigen presentation is only the first step in the activation of T cells, and subsequent co-stimulatory signals are important drivers of the resulting T cell response [66]. In this respect, differences between professional and nonprofessional APCs are apparent, for example, the absence of the co-stimulatory molecules CD80 and CD86 on most fibroblasts. These co-stimulatory molecules are present on professional APCs under physiological conditions, and this enables them to prime naïve T cells and stimulate differentiation to effector T cells. A large body of in vitro data suggest that the absence of these co-stimulatory molecules, together with PD-L1 expression, can lead to the induction of immunological tolerance by nonprofessional APCs. In contrast, inflammatory cues can cause CD80/86 upregulation, enabling T cell priming by nonprofessional APCs, as well as reinvigorating memory T cell responses. The exact inflammatory cues, besides IFNγ, that regulate the expression of co-stimulatory molecules on nonprofessional APCs are currently unknown, and it would be of great interest to identify the mediators and co-regulatory signals involved. In this regard, the identification of CAF-mediated PD-L2 and FASL signaling as mediators of tumor-specific T cell dysfunction [32] is a prime example of identifying the co-regulatory signals involved in nonprofessional APC-mediated immune tolerance. Particularly for the field of immuno-oncology, where tolerance induction of tumor-specific T cells by nonprofessional APCs is undesired, this is of considerable interest. Conversely, dampening these inflammatory mediators and associated co-regulatory signals in auto-immune diseases, such as diabetes, might help in restraining the ongoing harmful T cell response that can be potentiated by nonprofessional APCs.

There are currently several knowledge gaps regarding the in vivo effect of nonprofessional antigen presentation. First, detailed assessments of the type of inflammatory cues present in vivo, in both autoimmune diseases and oncology, are necessary to identify factors that modulate both the antigen presentation capacity, as well as the expression of co-stimulatory and co-inhibitory signals in nonprofessional APCs. Second, the type of antigen investigated in most studies introduces a potential bias for establishing the true effect of nonprofessional antigen presentation as, until now, mostly soluble antigens, such as peptides and proteins, have been investigated. Cell-associated antigens, also called particulate antigens, such as bacterial or apoptotic cell debris, are an important source of antigens in vivo and can be processed differently than soluble antigens [67]. Hence, the efficiency in uptake and processing can differ between the two antigen classes, and it is thus relevant to more thoroughly study the effects of particulate antigen presentation by nonprofessional APCs. Finally, with regard to immuno-oncology, it is largely unknown what the contribution of nonprofessional APC-mediated exogenous antigen presentation is to the functioning of tumor-specific T cells. This includes the effect of immune checkpoint inhibition on nonprofessional APC-mediated antigen presentation to T cells. In this regard, it is important to note that the previously held assumption that inhibitors of the PD1-PDL1 axis work solely by inhibiting T cell–tumor cell interactions has been proven false [68,69,70,71]. These studies showed that PD-L1 expression on hematopoietic stromal cells is also a determinant of success of checkpoint inhibitor therapy. It is currently unknown whether PD-L1 expression by nonhematopoietic stromal cell types, including the ones discussed in this review, also contributes to T cell suppression, and this is an important avenue to further investigate. Additionally, FASL and PD-L2 might also serve as potential targets for dampening nonprofessional APC-mediated T cell suppression, but their role in human disease needs to be further established.

In conclusion, solid evidence regarding the ability of nonprofessional APCs to present exogenous antigens and directly influence T cell function exists for a multitude of stromal cell types. The net effect of antigen presentation by stromal nonprofessional APC is dependent on the (tumor) micro-environment, and the inflammatory mediators and co-regulatory signals involved are rapidly being unraveled. Further research in the complex field of stromal immunobiology is warranted to enhance various immunotherapies by modulation of the T cell response through nonprofessional APCs.


**Article highlights**
Many stromal cell types can function as nonprofessional APC by modulating CD8+ or CD4+ T cell function, through cross-presentation and MHC class II-mediated presentation, respectivelyEffects of antigen presentation on T cell function differ and depend on several factors, such as local inflammation and expression of co-stimulatory and co-inhibitory molecules, including CD80/86 and PD-L1Nonprofessional APCs are involved in various pathologies such as cancer and auto-immune diseases

**Knowledge gaps**
The (inflammatory) mediators present in the micro-environment and their effect on exogenous antigen presentation by nonprofessional APCsHow efficiently do nonprofessional APCs present cell-associated antigens in comparison to soluble antigensThe contribution of nonprofessional APCs vs. professional APCs on the net effect of T cell function in vivo and the potential to modify nonprofessional APC–T cell interactions to treat disease


## Figures and Tables

**Figure 1 ijms-23-00137-f001:**
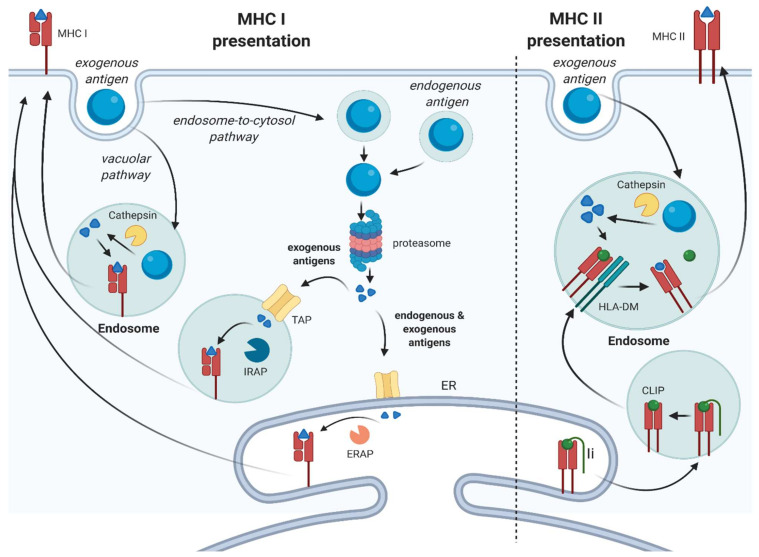
Exogenous antigen processing. MHC I presentation: in the endosome-to-cytosol pathway, exogenous soluble and particulate antigens are processed by the proteasome and enter either the ER or the endosome via TAP and are further processed by ERAP or IRAP, respectively. Endogenous antigens are also processed by the proteasome in the cytosol and enter the ER. Then, the antigens are loaded on the MHCI. In the vacuolar pathway, the antigen is processed by cathepsin S and is loaded onto the MHCI in an endosome. MHC II presentation: the antigen is processed by cathepsins in the endosome after internalization. The MHCII molecule is formed in the ER and Ii is cleaved in the endosome, which leaves CLIP in the binding pocket of MHCII. In the endosome, CLIP is removed by HLA-DM, so the antigen can bind. ER: endoplasmic reticulum; Ii: invariant chain; CLIP: class II-associated invariant chain peptide; TAP: transporter associated with antigen processing; IRAP: insulin-regulated aminopeptidase; ERAP: endoplasmic reticulum aminopeptidase; MHC: major histocompatibility complex.

**Figure 2 ijms-23-00137-f002:**
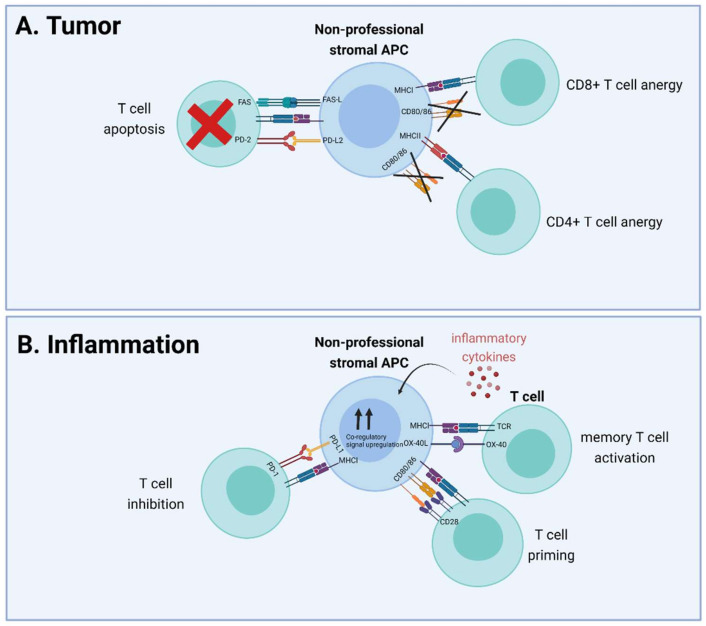
Effect of nonprofessional stromal APC antigen presentation on T cell function in tumor and inflammatory conditions. (**A**) CAFs reside in the tumor micro-environment, cross-present exogenous antigen cells to CD8+ T cells, and delete them in a PD-L2- and FAS-L-dependent manner, or cause anergy in T cells after antigen presentation by either MHCI or MHCII due to the lack of co-stimulatory molecules. (**B**) Inflammation results in the release of inflammatory cytokines that can cause upregulation of co-stimulatory (CD80, CD86, OX-40L) or co-inhibitory (PD-L1) molecules. Predominant upregulation of co-stimulatory signals enables T cell priming (CD28-CD80/86) interaction and memory T cell activation (OX-40L-OX-40) by nonprofessional stromal APCs, while upregulation of PD-L1 results in T cell inhibition. APC: antigen-presenting cell; PD-L: programmed death-ligand; TCR: T cell receptor; MHC: major histocompatibility complex; FASL: FAS ligand.

## Data Availability

Not applicable.

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
