# Peer review of "The ABCs of Antigen Presentation by Stromal Non-Professional Antigen-Presenting Cells"

_ijms, 2021, doi:10.3390/ijms23010137_

Round 1
Reviewer 1 Report
Authors should discuss/incorporate the below points in the manuscript
1. Authors should describe the role of non-professional antigen presenting cells in others diseases alongwith cancer.
2.Spell check is needed to correct the typo errors. e.g. MCH
3. How professional antigen presenting cells are superior over non-professional cells.
4. Can these non professional antigen presenting cell be modified to treat the diseases?
Author Response
Our point-to-point response to Reviewer 1 has been attached ('Reviewer 1 revision_final.docx').

Reviewer 2 Report
Harryvan et al. provide a survey over stromal cells being able to present antigens with and/or without co-stimulation. The former have a role in autoimmunity and possible antitumoral responses, the latter in thymus-independent tolerance due to the key mechanism of anergy. The review is comprehensively written, naming possible white spots and allowing to draw further hypotheses for further descriptive and mechanistic studies. I only have few minor points.
Minor issues:
- Line 62: The content of quotes 6,7 (reviews) is already mentioned in citation 4 (Joffre et al).
- Line 120: Here, “CAF” is mentioned for the first time (check together with line 169).
- Line 121: “…MHCII[23], have been first identified in pancreatic ductal adenocarcinoma (PDAC).” -> …MHCII, have been first identified in pancreatic ductal adenocarcinoma (PDAC)[23].
- Lines 179-189 and 190-192: Please briefly clarify where your data relies on in vivo, ex vivo and in vitro experiments.
- Line 213: “…were show..” -> were shown.
- Line 313: “…, potentiating cytotoxic T cell function.” Please add quote.
- Line 402: “…antigens5.” Please check quote.
Author Response
Our point-to-point response to Reviewer 1 has been attached ('Reviewer 2 revision_final.docx').
